# Geological Significance of Late Permian Magmatic Rocks in the Middle Section of the Ailaoshan Orogenic Belt, SW China: Constraints from Petrology, Geochemistry and Geochronology

Yaoyao Zhang [1], Kai Liu [1,*], Ye Wang [1,*], Da Zhang [2], Xuanxue Mo [2], Yuefei Deng [3], Tingxi Yu [1,2] and Zenan Zhao [4]

1 Laboratory of Deep Earth Science and Exploration Technology, Ministry of Natural Resources, Chinese Academy of Geological Sciences, Beijing 100037, China; zhangyy@cags.ac.cn (Y.Z.); yutx2020@163.com (T.Y.)
2 School of Earth Sciences and Resources, China University of Geosciences (Beijing), Beijing 100083, China; zhangda@cugb.edu.cn (D.Z.); moxx@cugb.edu.cn (X.M.)
3 CIGIS (CHINA) LIMITED, Beijing 100007, China; yuefeid@163.com
4 Hebei Regional Geological Survey Institute, Langfang 065000, China; zhaozenan1985@126.com
* Correspondence: liukai@cags.ac.cn (K.L.); ywang@cags.ac.cn (Y.W.)

**Abstract:** The Ailaoshan orogenic belt, located in the SE margin of the Qinghai–Tibet Plateau, is an important Paleo-Tethys suture zone in the eastern margin of the Sanjiang Tethys tectonic domain. The areas of Mojiang and Zhenyuan, located in the middle part of the Ailaoshan orogenic belt, are the key parts of the Ailaoshan Paleo-Tethys Ocean closure and collision orogeny. The rhyolites outcropped in the Mojiang area, and the granite porphyries outcropped in Zhenyuan area, are systematically studied for petrology, isotope geochemistry and geochronology. The Zircon U-Pb geochronology of rhyolites and granite porphyries give weighted average ages of 253.4 ± 4.2 Ma and 253.3 ± 2.0 Ma, respectively, both of which were formed in the late Permian period. The rhyolites belong to potassic calc-alkaline to subalkaline series. The patterns of the rare earth elements (REE) show a right-inclined seagull-type distribution, and the trace elements plot is right-inclined. The granite porphyries are high potassic calc-alkaline to subalkaline. The REE patterns show a right-inclined distribution, and the trace elements plot is right-inclined, which is consistent with the typical patterns observed in the crust. The peraluminous, highly differentiated and high ASI values suggest that rhyolites and granite porphyries are S-type granites. The zircon $\varepsilon Hf(t)$ of the rhyolites range from −7.22 to −0.72, and two-stage Hf zircon model ages are $(T_{DM}{}^{C})$ 1771–2352 Ma, indicating that the magma source area is mainly crust-derived. The zircon $\varepsilon Hf(t)$ of the granite porphyries range from −0.97 to 4.08, and two-stage Hf zircon model ages are $(T_{DM}{}^{C})$ 1336–1795 Ma, indicating that the magma is derived from a depleted mantle source and the partial melting of ancient crustal materials. The rhyolites and granite porphyries were possibly formed in the syn-collisional tectonic setting during the late Permian, and their ages limited the time of the final closure of the Ailaoshan Ocean and the initiation of collisional orogeny.

**Keywords:** rhyolites; granite porphyries; geochemistry; Lu-Hf isotopes; Ailaoshan orogenic belt

## 1. Introduction

The Sanjiang Paleo-Tethys Orogen is one of the important branches of the eastern Tethyan tectonic domain, and it preserves the records of the evolution of the Tethyan tectonic domain within Southeast Asia [1–5]. It is favored by researchers for its unique tectonic evolution history [6–13]. In the study of the Sanjiang Tethyan Orogen, the tectonic deformation of Meso-Tethys and Neo-Tethys tectonic events is apparent, while the Paleo-Tethys still has some key scientific problems, such as the closure time of the suture zone and the response of magmatic activity due to the superposition and complexity of the tectonic evolution and the limited research studies [14–18].

Tectonically, the Jinshajiang–Ailaoshan orogenic belt lies in the SE part of the San-jiang region, west of the South China Block [13,19–21], showing the geological boundary between the South China Block and the Indochina Block (Figure 1). Its special tectonic position indicates that this area is a key region for understanding the tectonic evolution and magmatic response of the closure of Ailaoshan Paleo-Tethys Ocean [17,18,22]. At present, there are many geochronological and geochemical reports on the collision-type magmatic rocks of the Jinshajiang orogenic belt. Zi et al. [23] found that the SHRIMP zircon U-Pb ages of the rhyolites in the Pantiange Formation were 245~237 Ma. Combined with Sr-Nd isotope, these authors found that the strata were deposited after the closure and orogeny of the Jinshajiang Ocean that separated the Indochina Block and the South China Block. Wang et al. [24] obtained the LA-ICP-MS zircon U-Pb ages of rhyolites within the bimodal volcanic rocks from the Deqin area between 249 and 247 Ma, suggesting that the Jinshajiang orogenic belt had entered the post-collisional extension period in the Early Triassic. Zhu et al. [25] measured the SIMS zircon U-Pb ages of granites in the Yangla area as 234~231 Ma, and whole-rock Sr-Nd-Pb isotopes and zircon Hf-O isotopes showed a later- or post-collision environment, concluding that the collision between the South China Block and the Indochina Block was completed before the Late Triassic. These studies enriched the data of Paleo-Tethys evolution and better constrained the closure time of the Jinshajiang Ocean [26,27]. In contrast, the evidence of magmatic rocks in the Ailaoshan orogenic belt that can limit the closure time of the Ailaoshan Ocean remains somewhat insufficient.

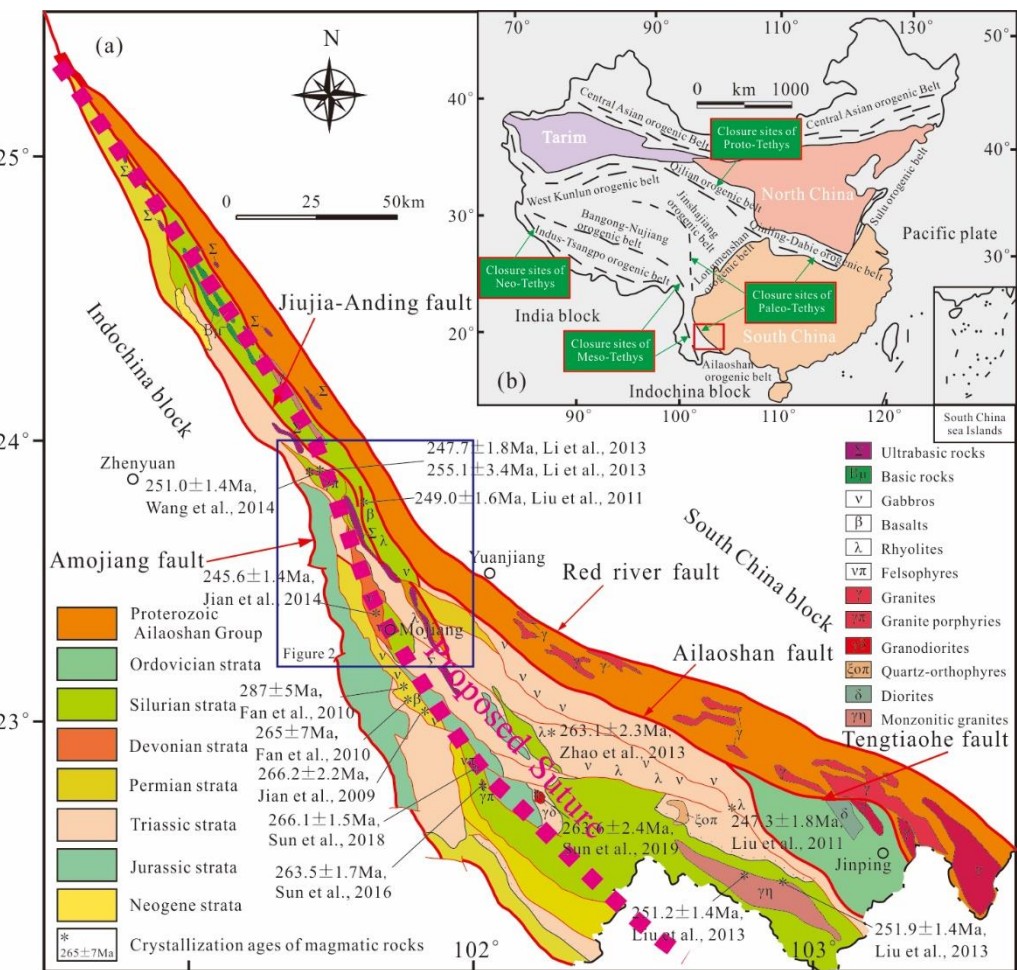

**Figure 1.** Temporal–spatial distribution of the magmatic rocks along the Ailaoshan orogenic belt (**a**) and the location of the study area in China's tectonic map showing Precambrian blocks surrounded by Phanerozoic orogenic belts ((**b**); modified from [28]).

Mojiang and Zhenyuan gold deposits are located in the middle section of Ailaoshan orogenic belt, and magmatic rocks develop in the mining district (Figure 2). Previous studies on this area mainly focused on the geological characteristics and genesis of the deposits, ignoring the chronological and tectonic significance of the magmatic rocks [29–41]. The rhyolites studied in this paper are from an outcrop in the Mojiang mining area, and the granite porphyries researched in this paper are from an outcrop in the Zhenyuan area. Through systematic analyses of their mineralogy, geochemistry, isotope chronology and Lu-Hf isotope characteristics, petrogenesis and tectonic evolution are discussed, and the closure time of the Ailaoshan Paleo-Tethys Ocean is constrained.

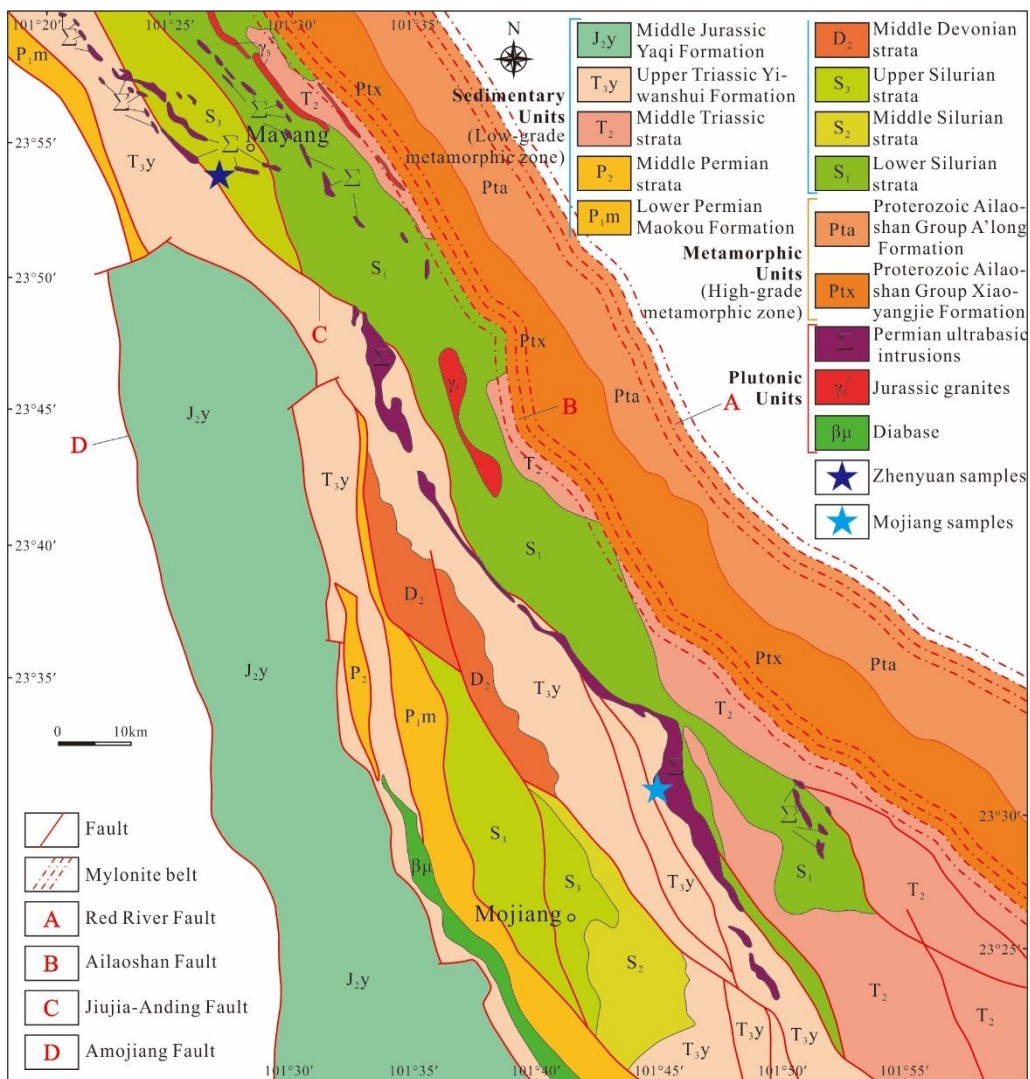

**Figure 2.** Geological sketch map of mining area (modified from [16]).

## 2. Geologic Setting

### 2.1. Geological Overview of the Region

The Ailaoshan orogenic belt, sandwiched between the Indochina Block in the W and the South China Block in the E, has undergone a complicated tectonic evolution, including compression collision, ocean basin expansion, plate subduction, thrust–nappe deformation and strike–slip shear, and forms the basic structural framework of four NW-trending regional faults (The Red River fault, the Ailaoshan fault, the Jiujia–Anding fault and the Amojiang fault) and two in-between metamorphic zones (the high-grade metamorphic zone and the low-grade metamorphic zone) [1,5,42].

The Proterozoic Ailaoshan Group (Pt) is exposed on the NE side, bracketed by the Ailaoshan fault in the southwest and the Red River fault in the northeast. The litostratigraphical units were intensively metamorphosed under long-term tectonic movement, resulting in migmatizaton. This group constitutes the crystalline basement of the Yangtze block. The Ailaoshan Group includes the Xiaoyangjie Formation (Pt*x*) and A'Long Formation (Pt*a*) [35,36]. The lithology of the lower section of Xiaoyangjie Formation primarily consists of two-mica schists and biotite granulites, and the upper section is mainly biotite plagioclase gneisses and biotite amphibolites. The lower section of the A'Long Formation consists of mainly amphibolite plagioclase gneisses, amphibolite granulites and diopside marbles, and the upper lithology primarily includes marbles, biotite amphibolite granulites and plagioclase amphibolites. The two formations show fault contact, and are a set of metamorphic rock series dominated by amphibolite facies.

Sandwiched between the Ailaoshan fault and the Jiujia–Mojiang fault, the Palaeozoic strata are exposed on the west slope of Ailaoshan and include Ordovician, Silurian, Devonian and Permian strata [38,39]. They are composed of clastic, carbonate and volcanic-sedimentary rocks affected by a low-grade metamorphism in greenschist facies conditions. The Ordovician outcrops lie in the southern part of the area and connect with the Ailaoshan Group through the fault, showing a parallel unconformity with the Silurian system. The Silurian system connects with the Ordovician system by parallel unconformities and displays angular unconformities with the Devonian system. The Silurian lithology is characterized by silty slates, quartz sandstones, siltstones, calcareous slates, limestones, dolomitic limestones, dolomites and shales. The Devonian strata are a set of shallow–deep marine clastic rocks and carbonate deposits, and its lithology is mainly sandstones, siltstones, shales, siliceous rocks, dolomitic limestones, biological limestones, etc. The Carboniferous strata are dominated by carbonates with a small number of coal-bearing strata and the local occurrence of basalts. The strata are mainly composed of limestones and dolomites. Permian outcropping mainly occurs in the NE side of the Jiujia–Mojiang fault zone and partly in the SW side of the Ailaoshan fault zone. It is dominated by neritic carbonate rocks, and the lithology is primarily characterized by quartz sandstones, silty shales, andesites, basalts, rhyodacites, andesite agglomerates, volcanic breccias, tuffs, etc. The Mesozoic strata are rarely exposed and mainly include shales, argillaceous limestones, siltstones, quartz sandstones and conglomerates. The Cenozoic strata are primarily composed of clastic rock deposits and are locally intercalated with carbonate deposits. The low-lying areas are covered with Quaternary rocks.

Regional magmatism features long-term and multi-period activities, lasting from Paleozoic to Cenozoic with outcrops of ultrabasic to alkaline rocks. The outcrop morphology, scale, and distribution characteristics of magmatic rocks are all restricted and controlled by the four regional faults (The Red River fault, the Ailaoshan fault, the Jiujia–Anding fault and the Amojiang fault). Ultrabasic rocks are mostly in the form of dikes, lenticles and bedrocks in the regional faults [17,43,44]. The scales generally range from tens to hundreds of meters with a few up to one thousand meters. The lithology are characterized with dunites, orthopyroxenite, augite-bearing peridotites and pyroxenites. Basic rocks are mostly distributed in the Jiujia–Anding abyssal fault zone, including gabbros and lamprophyres, which show a zonal distribution, trending mostly NW, in the form of veins and lenticles roughly distributed along the beds or along structural fracture zones and secondary structures. Their scales vary from a few centimeters at the narrow end to several meters and even up to tens of meters at the wide end. Their lithology includes lamprophyres, minettes and olivine–pyroxene minettes. The intermediate rocks are primarily present on the NE side of the Amojiang fault south of Ailaoshan. The largest rock bodies are the Taojiazhai and Qizanmi rocks, mainly composed of diorites. Acidic rocks are widely developed, mainly exposed on the NE and SW sides of the Ailaoshan fault, and distributed along the Jiujia–Anding fault. Alkaline rocks also occur extensively in the form of veins, lenticles and rock stocks. Some rock bodies are about 12 km in length and 2 km in width and are composed of syenites and orthoclase porphyries.



Regarding regional structures, the Ailaoshan region is located in the SE section of the orogenic zone where the Indian Block and the Eurasian Block converge and collide, resulting in strong tectonic movement where regional faults and secondary associated faults develop. The major regional faults include the Red River fault, Ailaoshan fault, Jiujia–Anding fault and Amojiang fault [8,15,36]. The Red River fault strikes 310–325°, and the northwest section dips north-eastward with a dip angle of 60–70°. It has compression and torsion characteristics with a left lateral strike–slip pattern in the early stage and right lateral strike–slip in the later stage. The Ailaoshan fault zone is the south-eastward extension of the Jinshajiang–Ailaoshan fault zone, striking 285–300° and dipping north-eastward with a dip angle of 30–70°. The dip angle is very steep at places. A large-scale mylonite belt is present with a width of 1–3 km, characteristic of reverse thrust nappe and strike–slip shear indicating strong transformation in the later period. The Jiujia–Anding fault zone extends in NW–SE direction with a dip angle of 45–80°and is characterized by multi-period and multi-stage activities, such as reverse thrust nappe and strike–slip shear. The Amojiang fault zone extends NW–SE and slightly protrudes to the southwest, dipping to the NE with a dip angle of 40–80°. The fault zone consists of multiple approximately parallel faults forming an imbricate reverse thrust structure in the section.

### 2.2. Geological Characteristics of Mining Area

The Mojiang gold deposits are located in the middle section of the Ailaoshan orogenic belt. Silurian, Triassic and Quaternary strata are exposed in the mining areas. The Silurian stratum is the oldest exposed strata in the mining area, consisting of three lithological members of the middle and lower Jinchang Formation. The lithology is primarily characterized by slates, quartzites, metamorphic siltstones and sandstones [45]. The rhyolites overlay the Silurian strata in an angular unconformity. The Triassic layer is composed of the Luma Formation and the Yiwanshui Formation. Outcrops of the Luma Formation, mudstones and sandstones, are on the W periphery of the mining area. The upper section of Yiwanshui Formation comprises siltstones and mudstones and its outcrops are located in the periphery of the mining area, while outcrops of the lower section are on the W part of the mining area characterized by mudstones and conglomerates. The Quaternary strata have a residual slope accumulation. Magmatic rocks are developed in the mining area and mainly include ultrabasic and acidic rocks. Ultrabasic rock forms a huge rock wall with narrow ends and an enlarged central part in a northwest–south reversed "S" shape on the plane. The rock body is 16 km long and 1.3–2 km wide with an area of about 20 km$^2$, intruding into the Silurian Jinchang Formation. Its lithology primarily features peridotites. Acidic rocks are mostly rhyolites, which overlie the Silurian strata in an angular unconformity and occur in the NNW direction (Figure 3a). The mining area is mainly connected by the Jiujia–Anding fault, which affects the occurrence of gold and nickel deposits. Folds and faults are relatively well-developed in the area. The Jinchang inverted anticline is the major fold structure of the mining area with the Jinchang Formation as the core structure and the Yiwanshui Formation as two wings. The Jinchang anticline axially aligns with the direction of the main structural lines of the area, spreading in the NW direction [45]. The fault structure mainly consists of three groups, trending in NW, NNW and NWW directions. The Jinchang fault is part of the middle section of the Jiujia–Anding abyssal fault. As the primary fault of the area, it runs through the entire mining area. Its overall strike is NNW dipping NEE with a dip angle of 45–80°. From S to N, the strike changes from NW to SN, and then to NW again with a wave-like curve along the way. As the main ore-controlling structure, the fault zone ranges from several hundred meters to more than two thousand meters in width [46].

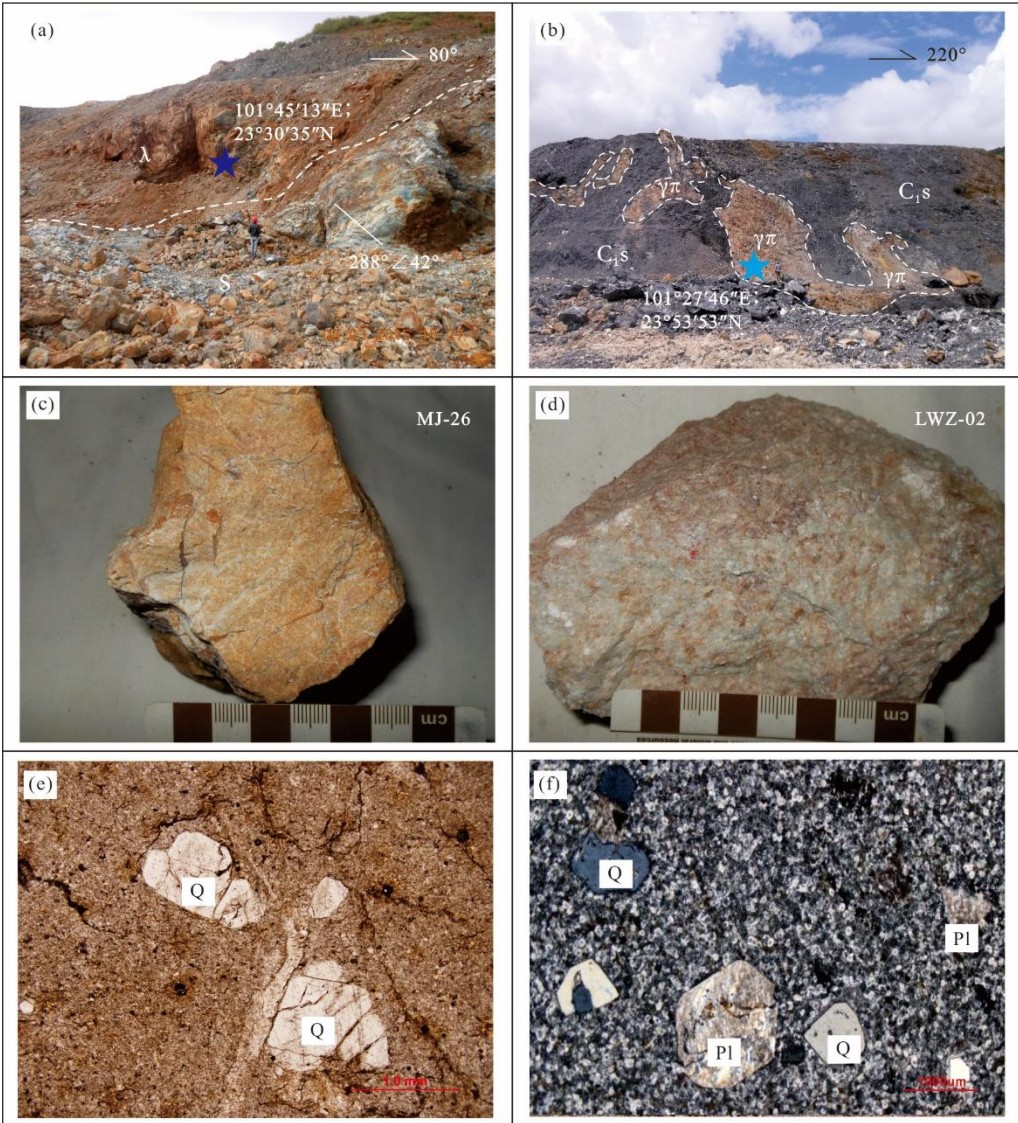

**Figure 3.** Field photos, hand specimens, and micrograph of the rhyolite in Mojiang gold deposits and the granite porphyry in Zhenyuan gold deposits. (**a**) Angular unconformity contact between rhyolite and Silurian strata; (**b**) Granite porphyry dike in Laowangzhai Mine; (**c**) Photograph of hand specimen of rhyolite (MJ-26) used for zircon U-Pb dating; (**d**) Photograph of hand specimen of granite porphyry (LWZ-02) used for zircon U-Pb dating; (**e**) Rhyolite phenocrysts are subrounded corroded quartz crystals; (**f**) Granite porphyry phenocrysts are sericitized feldspar and corroded quartz crystals. Q—quartz; Pl—plagioclase.

Zhenyuan gold deposits are also located in the middle section of the Ailaoshan orogenic belt. Late Paleozoic, Mesozoic and Cenozoic strata are exposed in the mining areas. The lower section of upper Devonian Kudumu Formation ($D_3k^1$) is mainly composed of siliceous slates and sericite siliceous slates, and the middle section ($D_3k^2$) mainly comprises carbonaceous argillaceous limestones. The upper section ($D_3k^3$) mainly contains metamorphic fine-grained quartz greywackes with a large number of quartz porphyry, granite porphyry and lamprophyre dike intrusions, as the main ore host strata of the Dongguolin ore section [47]. The Kudumu Formation and overlying Suoshan Formation show fault contact. The lower section of the lower Carboniferous Suoshan Formation ($C_1s^1$), include micritic limestones and carbonaceous calcareous slates. The upper section of the lower Carboniferous Suoshan Formation ($C_1s^2$) mainly involves carbonaceous calcareous slates, carbonaceous sandy sericite slates and metamorphic quartz greywackes. The lower

Permian Nazhuang Formation ($P_1n$) mainly consists of sericite phyllites, metamorphic quartz greywackes and calcareous slates. The upper Triassic Yiwanshui Formation ($T_3y$) is mainly composed of tuffaceous siltstones and detritus quartz greywackes. The Quaternary Formation is distributed in valley terraces, slopes, and swales, mainly in alluvial–diluvial and residual slope deposits. Magmatic rocks are developed in the mining area and distributed in groups, with outcrops ranging from ultrabasic rocks to intermediate-acid rocks, intruding into the Devonian and Carboniferous strata. Ultrabasic rocks are mainly olivine pyroxenite and augite peridotite, and intermediate-acid dikes are mainly granite porphyry (Figure 3b), quartz porphyry and granodiorite porphyry. The ages of the granite porphyry and quartz porphyry are Permian [17]. The deposit is located at the oblique junction of the NW-trending Jiujia–Anding fault zone and the nearly EW-trending Bankahe fault. The Neogene tectonic movement led to the reactivation of faults and the migration and precipitation of ore-forming fluids in Mojiang and Zhenyuan areas, and finally formed the gold deposits [1,13]. The NW-trending structure is dominant, followed by the EW-trending structure. The structure is characterized by multi-stage reverse thrust nappe and inherited ductile shear with a strong mylonitization. The NW-trending faults are relatively developed, most of which are the secondary faults of Jiujia–Anding fault zone, constituting the piggyback thrust belt with the tendency of NE, steep at the top and slow at the bottom. The EW-trending faults are conjugate structures derived from NW faults and are characterized by sinistral strike–slip shear.

## 3. Materials and Methods

### 3.1. Sample Collection and Features

Based on a comprehensive geological survey carried out in the field, eight fresh samples with weak alterations from the two mining areas were collected for geochemical analysis, two of which were also tested for U-Pb dating and a Hf isotope analysis of zircon. The sampling locations are shown in Figure 2.

Rhyolite: light grey or off-white, rhyolitic with a porphyritic texture (Figure 3c). The phenocryst/matrix ration is about 1/3. The phenocrysts include quartz, potassium feldspar, plagioclase, and biotite, mostly with a grain size of 0.2–1 mm. Quartz (10%) appears as anhedral grains with slight wavy extinction, and has subrounded corroded rims, with a matrix texture. Potassium feldspar (8%) forms subhedral and anhedral grains, partially angular and corrosion structures, developing transverse cracks and kaolinization along them. Plagioclase is subhedral–anhedral, develops polysynthetic twinning, zonal structure, and corrosion structure, amounting to approximately 6% of the content. Biotite is sheeted. The matrix has a microlithic texture (Figure 3e) and mineral composition of felsic (75%), magnetite, uranite, and zircon (1%).

Granite porphyry: light grey or off-white, blocky with porphyritic texture (Figure 3d). The phenocryst/matrix ration is about 1/4. The phenocrysts include potassium feldspar, plagioclase, quartz, and biotite, mostly with a grain size of 0.2–1 mm. Potassium feldspar (10%) appears as subhedral and anhedral grains, inordinately sericitized and kaolinized, mostly pseudomorphous. Plagioclase (5%) is present as subhedral and anhedral grains. The quartz (5%) appears as subhedral grains with slight wavy extinction. Biotite is present as flakes altered by chlorite and carbonate, with a content of 2%–4%. The matrix has a microlithic texture (Figure 3f), and its mineral composition is felsic (~80%) magnetite, uranite, and zircon (1%).

### 3.2. Experimental Methods

The chemical component analysis of major elements, trace elements, and REE of the samples was completed at the Laboratory of Institute of Geophysical and Geochemical Exploration, Chinese Academy of Geological Sciences. The major elements were determined by XRF method, according to standard GB/B14506.28-1993. The $H_2O^+$ analysis follows standard GB/T14506.2-1993, and the loss on ignition standard was LY/T1253-1999, with an analysis accuracy of 5%. REE and some trace elements (Hf, Th, Ta, U) were tested

with inductively coupled plasma mass spectrometry (ICP-MS), according to standard DZ/T0223-2001. Trace elements (Nb, Zr, Ga, Sr, Ba, and Rb) were determined using an X-ray fluorescence spectrometer (Model 2100), following standard JY/T016-1996. The analysis accuracy for most elements was $10^{-8}$, with a few at the levels of $10^{-6}$ (Zr and Ba) and $10^{-7}$ (Hf and Nb), and the relative standard deviation was less than 5%.

Zircon separation was conducted in the Regional Geological Survey Institute of Hebei Province. The samples were processed through conventional comminution, magnetic separation, and gravity separation to obtain a high-purity heavy mineral concentrate; zircons were further manually selected using binoculars to achieve a purity of over 99%. Target samples were prepared by attaching selected zircon grains to epoxy resin, and then exposing the surface through grinding and polishing. The analysis of transmission, reflection and cathodoluminescence (CL) of zircon was conducted by Beijing Zirconia Pilot Technology Co., Ltd., Beijing, China, to identify the texture and structure of the inner growth layer of zircon. Then, the samples were sent to the Isotope Laboratory of Tianjin Geological Survey Centre, China Geological Survey, to complete the dating analysis.

Zircon dating involved U-Pb isotope analysis of zircon with laser ablation multi-collector inductively coupled plasma mass spectrometry (LA-MC-ICP-MS). The zircon U-Pb analysis was conducted using a 193 nm laser to perform zircon ablation, with laser ablation beam diameter at 35 μm, laser energy density at 13–14 J·cm$^{-2}$, and frequency of 8–10 Hz. Helium was used as the carrier gas to transport analytes to NEPTUNE. U-Pb isotopes with different masses can be simultaneously collected using dynamic zooming expanded dispersion. Plesovice (age of $337 \pm 0.37$ Ma) [48] was selected as the external standard for the calculation of Pb, U, and Th content of zircon samples, and the ComPbCorr#3.17 calibration procedure [49] was used for common lead calibration. The collected data were processed using the ICP-MS DataCal program and Isoplot program. Uncertainties in individual analyses are presented here as $\pm 1\sigma$. Concordia diagrams and relevant graphs were plotted with a 95% confidence. Detailed instrument operating conditions and data processing methods are described in Liu et al. [50].

Zircon Hf isotope analysis was also conducted using laser ablation inductively coupled plasma mass spectrometer (LA-ICP-MS) in the Isotope Laboratory of Tianjin Geological Survey Centre, China Geological Survey. The LA-ICP-MS laser ablation system is a NewWave UP193FX 193 nm ArF excimer system. The laser is an ATL product (Germany), and the ICP-MS is Agilent 7500a. The laser wavelength was 193 nm, the pulse width was <4 ns, and the beam spot diameter was 35 μm. Zircon standard sample 91,500 ($^{176}$Hf/$^{177}$Hf $= 0.282308 \pm 12(2\sigma)$) was used as the external standard for matrix calibration [51]. In the calculation of Hf mantle model age, the current value of $^{176}$Hf/$^{177}$Hf of the depleted mantle was 0.28325, and the current value of $^{176}$Lu/$^{177}$Hf of the depleted mantle was 0.0384 [52]. The mean crustal $^{176}$Lu/$^{177}$Hf was 0.015 for the crustal model age calculation [53].

### 3.3. Results

#### 3.3.1. Major, Rare Earth and Trace Elements

The rhyolites in this work contained 81.16%–82.62% $SiO_2$. The contents of $K_2O$ (2.54% to 3.25%) and $Al_2O_3$ (11.20% to 12.60%) were relatively stable, and the contents of $Na_2O$ (0.08%) and CaO (0.05%–0.12%), which were very low, were consistent with the observation that the phenocrysts are mainly quartz and K-feldspar. The aluminum saturation index A/CNK was 3.37–3.74 (Supplementary Table S1), which was much higher than that of normal acidic rocks and was related to the low content of CaO and $Na_2O$, indicating that the rocks had strong peraluminous properties. The contents of $TiO_2$, FeO$^t$, MnO, MgO and $P_2O_5$ were 0.01%, 0.41%–0.70%, 0.01%, 0.10%–0.55% and 0.06%–0.07%, respectively. We collected and sorted volcanic rocks data with similar geological significance by region to classify rock types and distinguish the aluminum saturation index [43,54]. In the diagram of $SiO_2$-$K_2O$ + $Na_2O$ (Figure 4a), the sample data are plotted in the rhyolite field and show subalkaline characteristics. In the $SiO_2$-$K_2O$ diagram (Figure 4c), the data are plotted in

the calc-alkaline range. The rock differentiation index (DI) is in the range from 88.5 to 89.9, showing a relatively high degree of differentiation [55].

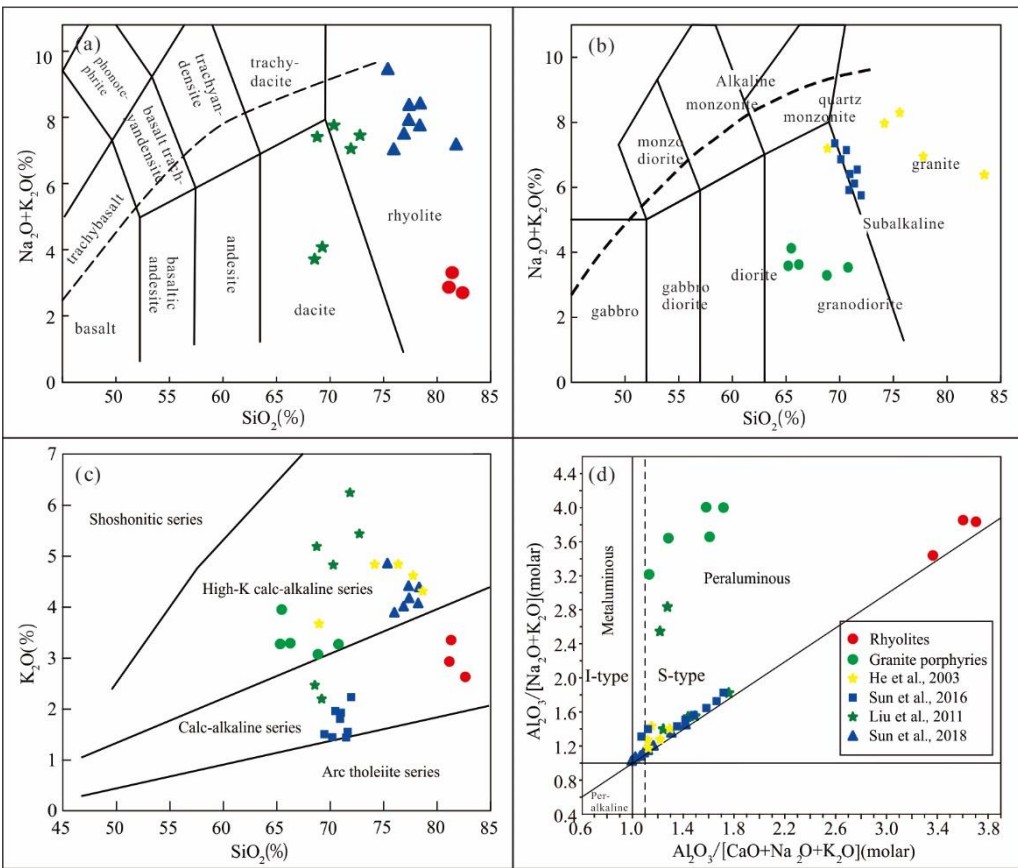

**Figure 4.** Classification and discriminant diagrams of magmatic rock. (**a**) $SiO_2$ vs. $K_2O + Na_2O$ plot of volcanic rocks [56]; (**b**) $SiO_2$ vs. $K_2O + Na_2O$ plot of intrusive rocks [56]; (**c**) $K_2O$ vs. $SiO_2$ plot [56]; (**d**) A/NK-A/CNK plot [57]. The samples of He et al. [30] are from the Zhenyuan gold deposits. The location and lithology of other samples from the references are shown in Figure 1.

The granite porphyries are intermediate-acid rocks with a $SiO_2$ content of 65.30%–70.79%. Their $Al_2O_3$ content was 13.42%–16.40%, and total alkali content ($K_2O + Na_2O$) was 3.28%–4.12% (Supplementary Table S1). In the TAS diagram (Figure 4b), the sample data of granite porphyries exhibited granodiorite, showing subalkaline characteristics. In the diagram of $SiO_2$-$K_2O$ (Figure 4c), the sample data of granite porphyries were deposited in the high-K calc-alkaline region. The rock differentiation index (DI) was 65.9–74.7, and the phenocryst contained dark minerals such as biotite, indicating that the rock was characterized by high differentiation. The aluminum saturation index A/CNK was 1.12–1.72, indicating that the rock was strongly peraluminous.

The REE content in the rhyolite was relatively low, with ΣREE content from 17.4 ppm to 45.8 ppm, whereas the LREE content varied from 11.1 ppm to 38.8 ppm, and HREE content ranged from 6.35 ppm to 7.87 ppm. The (LREE/HREE) was 1.75–5.50, and $(La/Yb)_N$ is 1.79–9.36. The REE patterns exhibited a right-inclined seagull-type distribution (Figure 5a) with negative Eu anomalies (Eu/Eu* = 0.42–0.52). The primitive mantle normalized trace element spider diagram of the rhyolite was tilted to the right (Figure 5b). High-field-strength elements (HFSEs), such as Ta and Nb, exhibited strong negative anomalies, while large-ion lithophile elements (LILEs), including Rb and U, showed positive anomalies. In addition, Ba and Sr in the three samples displayed noticeable negative anomalies, which could be related to the separation of plagioclase and amphibolite during the partial melting or crystal differentiation of magma [17].

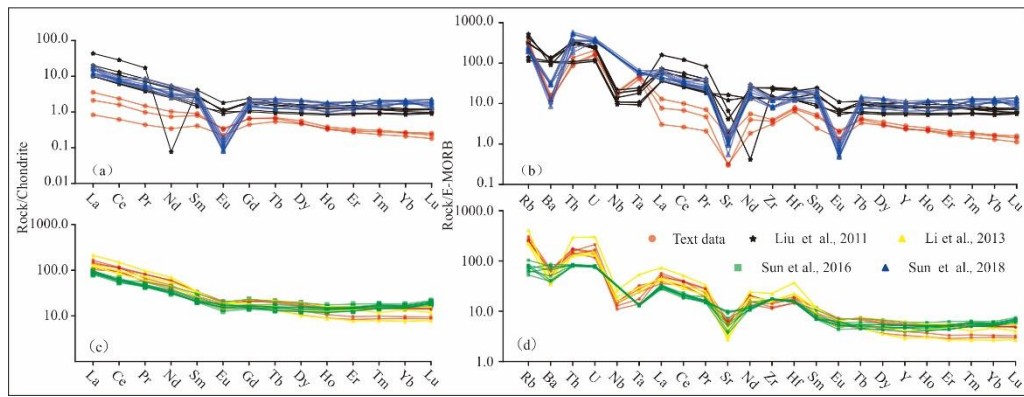

**Figure 5.** Chondrite-normalized REE and E-MORB-normalized trace-element spider patterns. (**a**) Chondrite-normalized REE patterns of volcanic rocks; (**b**) E-MORB-normalized trace-element spider patterns of volcanic rocks; (**c**) Chondrite-normalized REE patterns of intrusive rocks; (**d**) E-MORB-normalized trace-element spider patterns of intrusive rocks (normalizing factors are from [58]; Chondrite and E-MORB compositions are from [59]).

The REE content in granite porphyry is relatively low, with ΣREE content from 123 ppm to 168 ppm, where the LREE content is from 112 ppm to 152 ppm, and HREE content is from 10.1 ppm to 16.3 ppm. The ratio of light to heavy rare earths LREE/HREE is 6.90–13.4, and $(La/Yb)_N$ is 6.53–14.7. The REE patterns exhibit a right-inclined distribution (Figure 5c) with weak negative Eu anomalies (Eu/Eu* = 0.70–0.78). The primitive mantle-normalized trace-element spider diagram of the granite porphyry is tilted to the right (Figure 5d). High-field-strength elements (HFSEs) such as Nb and Ta exhibit strong negative anomalies, while large-ion lithophile elements (LILEs), including Rb, U, and La showed positive anomalies, with characteristics of mantle-derived rocks of a subduction zone.

### 3.3.2. LA-ICP-MS Zircon U-Pb Ages

In order to determine the crystallization age of the rhyolite in the Mojiang gold deposit, we selected the least altered sample (MJ26) for zircon U-Pb dating. The zircons had euhedral forms. Part of the crystal edges and apexes were eroded, showing a subcircular shape with particle sizes of 60–100 μm. The cathodoluminescence (CL) images clearly show the development of typical magmatic oscillatory zoning of zircon without regenerative metamorphic zircon edges (Figure 6a). Zircons without fissures and inclusions were selected for dating analysis, and spots were marked at the developed oscillatory zone, obtaining 15 sets of data. The content of Th and U varied from 124.23 ppm to 19,135.37 ppm and from 269.76 ppm to 61,760.92 ppm. Th/U ranged from 0.10 to 0.84, greater than 0.1, indicating typical magmatic zircons [60–62], and were consistent with the characteristics of magmatic oscillatory belts in zircon CL images. One zircon $^{206}Pb/^{238}U$ age was 480 Ma, indicating a relatively old inheritance age, which might be related to a tectonic thermal event of the early Paleozoic age. The other 14 zircon $^{206}Pb/^{238}U$ ages ranged from 235.2 to 287.5 Ma, and 10 of them were 246 ± 2.5 to 263 ± 3.1 Ma, with a weighted average age of 253.4 ± 4.2 Ma (MSWD = 4.3) (Figure 6b).

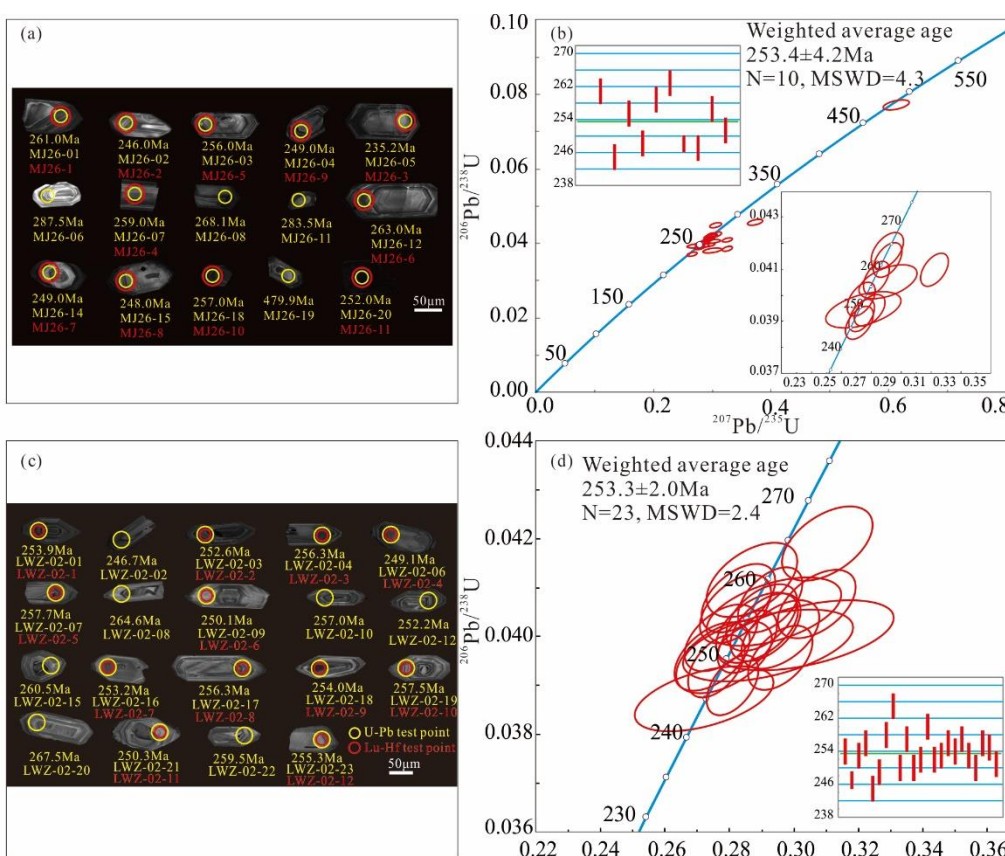

**Figure 6.** (**a**) Cathodoluminescence (CL) images of zircons of the rhyolite (MJ-26); (**b**) LA-ICP-MS zircon U-Pb concordia diagram of the rhyolite (MJ-26); (**c**) Cathodoluminescence (CL) images of zircons of the granite porphyry (LWZ-02); (**d**) LA-ICP-MS zircon U-Pb concordia diagram of the granite porphyry (LWZ-02).

Zircons from granitic porphyry have a fair euhedral form, showing a subcircular shape, with a length of 60μm~140μm and a length–width ratio of 1.5:1~3:1. Cathodoluminescence images show the development of oscillatory belts in zircons (Figure 6c), indicating the origin of magma [63]. Zircon Th content was in the range from 100.60 ppm to 298.58 ppm, and U content was 218.33 ppm–741.55 ppm. Th/U ranged from 0.10 to 0.84, greater than 0.1, indicating the magmatic origin of zircons [61,62] (Supplementary Table S2). In addition to the large deviation of test point 12, the other 23 zircon $^{206}$Pb/$^{238}$U ages ranged from 246.7 to 264.6 Ma, with a weighted average age of 253.3 ± 2.0 Ma (MSWD = 2.4), indicating that the hypabyssal rock was formed in the late Permian (Figure 6d).

### 3.3.3. Zircon Lu-Hf Isotopic Compositions

The Lu-Hf isotope analysis of zircons from the rhyolite was carried out at and around the spots where the U-Pb dating test was performed. Results are shown in Supplementary Table S3. The ratio of $^{176}$Lu/$^{177}$Hf is 0.000149–0.001125, less than 0.002, indicating that only a small amount of radiogenic Hf accumulated after the forming of zircon. Therefore, the initial $^{176}$Hf/$^{177}$Hf ratio can be used as the Hf isotopic composition during forming [64]. The $^{176}$Hf/$^{177}$Hf initial ratio and $\varepsilon$Hf($t$) values were calculated from the same zircon U-Pb age data. The two-stage model age ($T_{DM}{}^C$) was calculated based on the depleted mantle [52]. The Hf isotope ratio $^{176}$Yb/$^{177}$Hf was 0.004561–0.030908, the ratio $^{176}$Hf/$^{177}$Hf was 0.282419–0.282599, $f_{Lu/Hf}$ was between −1.00 and −0.97, and zircon $\varepsilon$Hf($t$) ranged from −7.22 to −0.72 (Figure 7a,b). The single-stage Hf zircon model age ($T_{DM}$) was 919–1183 Ma, and two-stage Hf zircon model ages ($T_{DM}{}^C$) 1771–2352 Ma were concentrated at 2.2 Ga.

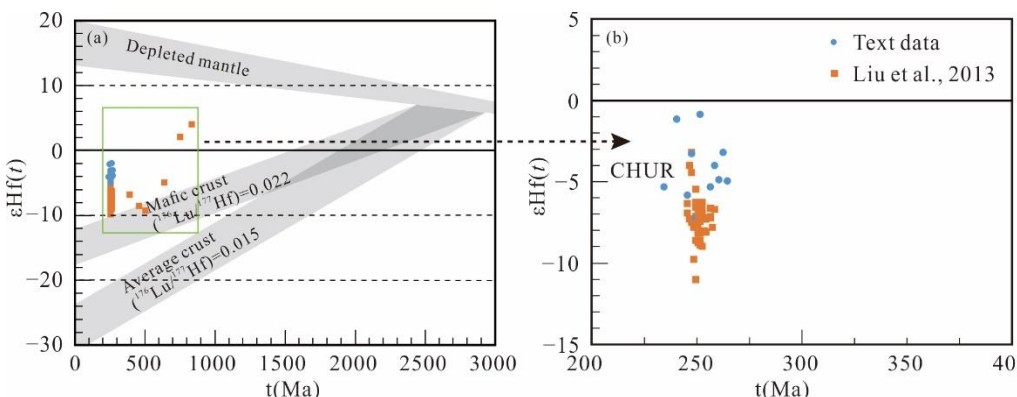

**Figure 7.** εHf(*t*) versus age diagram (**a,b**) (Base image according to [64]). The samples from Liu et al. [65] are volcanic rocks along the Ailaoshan orogenic belt.

Lu-Hf isotopic data of 12 zircons from sample LWZ-02 (granite porphyry) are shown in Supplementary Table S3. $f_{Lu/Hf}$ ratios ranged from −0.92 to −0.96, and εHf(*t*) varied from −0.97 to +4.08 with an average value of +1.21 (Figure 8a,b). The variation from negative value to positive values indicated that the rocks could be composed of heterogeneous zircons. Zircon single-stage Hf model ages ($T_{DM}$) ranged from 739 Ma to 945 Ma, and two-stage Hf model ages ($T_{DM}^{C}$) ranged from 1336 Ma to 1795 Ma.

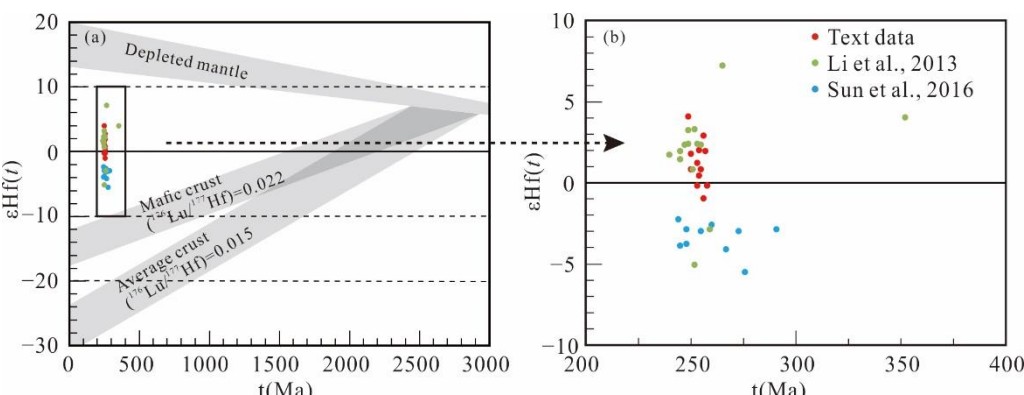

**Figure 8.** εHf(*t*) versus age diagram (**a,b**) (Base image according to [64]). The samples from Li et al. [17] and Sun et al. [44] are granite porphyries not far from the study area along the Ailaoshan orogenic belt.

## 4. Discussion

### 4.1. Rock Crystallization Ages

The study of rhyolite unit crystallization age in Mojiang gold deposits was lacking in this area. The technicians of the Mojiang Mining Company proposed that the rocks were formed in the Jurassic period based on their own experiences. Chen et al. [33] concluded that the rhyolites in Mojiang gold deposits were formed in the Jurassic period (180.3 ± 1.6 Ma, whole rock K-Ar method). There were few reports of Jurassic magmatic rocks in the region, and the whole-rock K-Ar method had a lower measurement accuracy, so we doubted the validity of this age. Due to its high physical and chemical stability [66], zircons have become ideal objects for determining the crystallization ages of magmatic rocks [67]. Zircons that were intact and euhedral (without cracks and inclusions) were selected and marked at the places where the oscillatory zone developed. The zircons of rhyolites showed a good correlation between Th and U, indicating typical magmatic zircons [68]. 10 analyses of rhyolite zircons yielded a weighted average age of 253.4 ± 4.2 Ma, suggesting that they were formed in the Late Permian. No Permian rocks were previously reported in the Mojiang area. This is the first time that we obtained a late Permian age for rhyolites, which also associated magmatic response with Ailaoshan Paleo-Tethys orogeny.

The zircon $^{206}Pb/^{238}U$ ages of granite porphyries obtained by Li et al. [17] in the Zhenyuan gold mine were 251.7 ± 2.1 Ma (N = 19, MSWD = 1.4) and 247.7 Ma ± 1.8 Ma (N = 20, MSWD = 1.2), and the samples were collected from veins about 1 m wide in Donggualin mining area from the Zhenyuan deposits. The zircon $^{206}Pb/^{238}U$ ages of quartz porphyries obtained by Li et al. [17] from Laowangzhai mining area of the Zhenyuan gold deposit were 255.1 ± 3.4 Ma (N = 13, MSWD = 2.9). By contrast, this quartz porphyry age had fewer analysis points and a larger margin of error. In addition, from the microscopic characteristics, the quartz porphyries described by Li et al. [17] should actually be granite porphyries. In this paper, the granite porphyry samples used in the experiment were collected from a rock wall about 15 m wide from the Laowangzhai mining area, where Zhenyuan gold deposits were found, which could better reflect the true crystallization age of the rocks. Although these ages were within the error range, we suggested that the crystallization ages of granite porphyries obtained by Li et al. [17] were a little younger, and the crystallization ages of granite porphyries are about 255 Ma. Combined with the previous research results [11,12,15–18,65], it is suggested that the Ailaoshan orogenic belt experienced intense magmatic activity in the late Permian, and such intensive and concentrated volcano responses may reflect that they were formed by the same tectonic thermal event.

### 4.2. Petrogenesis and Source Area Characteristics

The rhyolites of Mojiang gold deposits are characterized by high $SiO_2$, high Rb/Sr, high DI, low Zr/Hf, high differentiated granite [69], and low Co, indicating that the rocks had experienced crystallization differentiation. A/CNK ranged from 3.37 to 3.74 (Figure 4d), indicating that it was a strongly peraluminous and highly differentiated rock. Therefore, the rhyolite was regarded as S-type granite. There are two possible origins of peraluminous rocks: (1) partially melting and crystallization of lower crust or mantle-derived magma [70]; (2) partially melting of metamorphosed sedimentary rocks, such as clay-rich metamorphosed argillaceous rocks or clay-poor metamorphosed sandy rocks [71]. According to the geochemical characteristics of Rb, Ba, and Sr, Rb/Ba and Rb/Sr systems can reflect the composition of source rocks [72]. In the Rb/Ba-Rb/Sr diagram (Figure 9), most of the sample points of the volcanic rocks were plotted in the clay-rich source rock area, indicating that similar magmatic activities should have occurred under the tectonic background of the thickened crust and the result of melting in the crustal provenance. The rhyolite in the Mojiang gold deposit might be the product of the partial melting of sedimentary rocks (mainly argillaceous rocks) [71]. In the process of partial melting, the Nb/Ta value changed very little without the addition of foreign materials, and the Nb/Ta value of homologous magma remained the same [73,74]. The Nb/Ta value of the rhyolite ranged from 6.39 to 7.05, which was slightly smaller than the continental crust ratio (10) [75], indicating that the magma source area was crust-derived material. The enrichment of large ion lithophile elements and the depletion of high-field-strength elements such as Ta and Ti indicated that the magma forming these rocks mainly came from the crust. Zircon εHf (*t*) ranged from −7.22 to −0.72, also indicating that the magmatic source area was crustal material. Zircon εHf (*t*) and two-stage model age ($T_{DM}^C$) histograms showed a narrow tower distribution, suggesting a single source of magma.

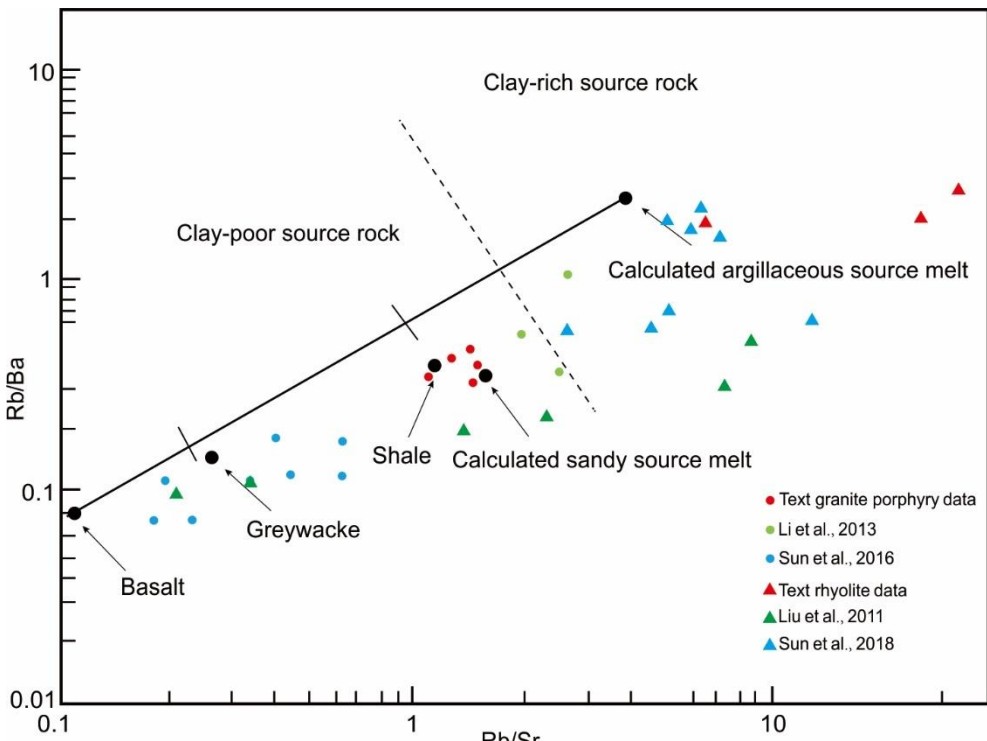

**Figure 9.** Rb/Ba vs. Rb/Sr diagram for magmatic rocks (base image according to [76]). The samples from Li et al. [17] and Sun et al. [44] are granite porphyries not far from the study area along the Ailaoshan orogenic belt. The samples from Liu et al. [54] and Sun et al. [43] are volcanic rocks not far from the study area along the Ailaoshan orogenic belt.

The zircon $f_{Lu/Hf}$ of rhyolite ranged from $-1.00$ to $-0.97$ (Supplementary Table S3), which was significantly smaller than that of continental crust ($-0.72$, [77]). Therefore, the two-stage model age can truly reflect the time when the source material was extracted from the depleted mantle (that is, the average age of the source material remaining in the crust) [78]. Zircon single-stage Hf model ages ($T_{DM}$) ranged from 919 Ma to 1183 Ma, and two-stage Hf model ages ($T_{DM}^C$) ranged from 1771 Ma to 2351 Ma. Zhai et al. [79] calculated the Sm-Nd isochron age of the amphibolite in the Ailaoshan group as $1367 \pm 46$ Ma, and according to the model age, the crystallization age of the amphibolite could be 1600 Ma~2000 Ma. Zhong [15] and Zhu et al. [80] found that Nd isotope model ages of the Precambrian deep metamorphic rock series (Ailaoshan Group, etc.) in the Ailaoshan orogenic belt were concentrated at 1600~1900 Ma. Combined with the Hf isotopic characteristics of zircons, we suggest that the ancient crustal materials may come from the Ailaoshan Group. Due to the lack of high $\varepsilon$Hf ($t$) values representing mantle source characteristics, it was considered that the mantle material did not participate in the forming of the magmatic rocks, but the mantle magma likely provided the heat source for the forming of the rhyolite.

No dark alkaline minerals were found in granite porphyries under microscope, and the chondrite-normalized REE pattern was right-leaning without an obvious negative Eu anomaly, which was clearly inconsistent with the characteristics of A-type granite [76]. A high Ga/Al value was one of the main characteristics of A-type granite, but the $10,000 \times$ Ga/Al value of granite porphyry was 1.83~1.98, which was significantly lower than 2.5 [72]. The $SiO_2$ content of granite porphyry in the study area was 65.30%~70.79%, and the magma was from the melting of $SiO_2$-rich source rocks. This narrow composition range was an important difference between S-type granites and I-type granites. In the A/NK-A/CNK discriminant diagram, data points were plotted on the S-type side (Figure 4d). In the CIPW standard mineral calculation, C (standard corundum molecule) ranged from 2.06 to 7.68 without a diopside. In summary, the granite porphyry was regarded as S-type granite [72].

S-type granites were mostly A/CNK > 1.1 generated after partial melting of sedimentary rocks [71]. In the Rb/Ba-Rb/Sr diagram (Figure 9), the sample points fell into the clay-poor source rock region, which was consistent with the sandy source melt calculated in the experiment. The above analysis indicated that the source rocks of the strong peraluminous granites in the study area were mainly sandy rocks, and argillaceous rocks might play a secondary role. The Nb/Ta value of granite porphyry had a wide range (7.94~12.68), indicating that the primary magma of granite porphyry was mainly crust-derived material and partly mantle-derived magma. Zircon $\varepsilon$Hf $(t)$ ranged from $-0.97$ to $4.08$, and was plotted between the depleted mantle evolution line and the crustal evolution line, suggesting that the granite porphyry magma source area was a mixture of depleted mantle magma source and ancient crustal provenance. The zircon $\varepsilon$Hf $(t)$ and two-stage model age $(T_{DM}{}^{C})$ histograms showed bimodal distribution patterns, indicating that the magmatic region was not a single source. The zircon $f_{Lu/Hf}$ of the granite porphyry ranged from $-0.92$ to $-0.96$, the single-stage Hf model ages $(T_{DM})$ ranged from 739 Ma to 945 Ma, and the two-stage Hf model ages $(T_{DM}{}^{C})$ ranged from 1336 to 1795 Ma. Combined with petrogeochemistry and zircon Hf isotope characteristics, we suspected that the source rock of granite porphyries in Zhenyuan gold deposit might be metamorphic sandstones of the Ailaoshan Group, and partly influenced by mantle-derived materials.

*4.3. Tectonic Significance*

High-field-strength elements (HFSE), such as Nb, Ta, Th, Zr, Hf, and HREEs, which are generally not impacted by hydrothermal alterations and metamorphism below the amphibolite facies, can effectively distinguish the tectonic environment of rocks [81]. Rb, Y (Yb), and Nb (Ta) are the most effective markers for distinguishing the most common types, such as ocean ridge granite (ORG), volcanic arc granite (VAG), within-plate granite (WPG) and syn-collision granite (Syn-COLG) [82]. In the Rb-Y + Nb tectonic discrimination diagram (Figure 10a), the sample points were projected at the boundary between the syn-collision and the volcanic arc area [82]. In the Rb-Yb + Ta tectonic discrimination diagram (Figure 10b), the sample data were projected at the boundary between the volcanic arc and the syn-collision area, and partly in the intraplate area. The sample points in the Nb-Y tectonic discrimination diagram (Figure 10c) were deposited in the volcanic arc and the syn-collision structure environment. In the Ta-Yb tectonic discrimination diagram (Figure 10d), the sample data were located close to the border of the syn-collisional and the volcanic arc environment, and partly intraplate area. This showed that the rock was formed in the tectonic environment of a continent–continent collision.

In recent years, the increasing research on the Ailaoshan orogenic belt has provided favorable conditions for a comprehensive study of the closure time of the Ailaoshan Paleo-Tethys Ocean. The Triassic Yiwanshui Formation was widely exposed in the Ailaoshan orogenic belt. Zhong [15] suggested that the collision occurred in the Middle Triassic according to the upper Triassic Yiwanshui Formation angular unconformity to the Ailaoshan ophiolitic melange. Jian et al. [11,12] obtained an age of $266.2 \pm 2.2$ Ma through the SHRIMP zircon U-Pb dating of basalts in the Yaxuanqiao area, indicating an island arc genesis of volcanic rocks. Sun et al. [43] obtained a zircon U-Pb age of $266.1 \pm 1.5$ Ma of felsophyre in Fengbieshan (in the south of Mojiang County), indicating that the rocks were formed in the post-collision intraplate tectonic environment. Fan et al. [16] measured the age of basaltic andesites in Yaxuanqiao area as $265 \pm 7$ Ma using SHRIMP U-Pb method, and found that the rocks had arc to back-arc characteristics. According to Liu et al. [54], the zircon SHRIMP U-Pb age of the Luchun rhyolite was $247.3 \pm 1.8$ Ma, suggesting that the area had entered the transition stage from mature island arc to continental collision at that time. Zhao et al. [83] carried out research on the Niangzong rocks in the NW of Jinping County, and considered that the forming time of the rocks to be about $263.1 \pm 2.3$ Ma, which was characterized by a collision zone and volcanic arc environment. Sun et al. [44] obtained a dating result of $263.5 \pm 1.7$ Ma from monzonite porphyry in the Niuzhi area (in the south of Mojiang County), indicating that the porphyry was formed in the tectonic environment of

island arc to continental collision or continental arc collision. The granodiorite in Mayu (in the south of Mojiang County) had a zircon U-Pb age of 263.6 ± 2.4 Ma; Sun et al. [84] suggested that it was formed in a post-collision intraplate tectonic environment. Li et al. [17] measured quartz porphyry and granite porphyry in the Zhenyuan area and obtained crystallization ages of 255.1 ± 3.4 Ma and 247.7 ± 1.8 Ma, respectively. Geochemical characteristics reflected that those granites were formed under the background of syn-collision structure. Liu et al. [65] measured the ages of Xin 'anzhai granite (in the west of Jinping County) as 251.9 ± 1.4 Ma and 251.2 ± 1.4 Ma and proposed that the rock was formed in the transition from island arc to continental collision or syn-collision tectonic environment. Liu et al. [65] revealed the existence of a new metamorphic event of high-pressure granulite facies in Ailaoshan orogenic belt between 249~230 Ma, which we suggested was related to the late Permian–early Triassic continental collision. Xu et al. [18] reported the detrital zircon U-Pb ages and Hf isotopic characteristics of Triassic sedimentary rocks on both sides of the Ailaoshan orogenic belt. The study showed that the Middle Triassic sediments on both sides of the suture line had clear similarities, indicating that that a continental collision occurred before the middle Triassic (Figure 11a). It could be seen that the ages of these magmatic rocks were concentrated in 265–245 Ma as a magmatic response to the continent–continent collision stage of the Paleo-Tethys orogeny, which was close to the closure time of the Ailaoshan Ocean.

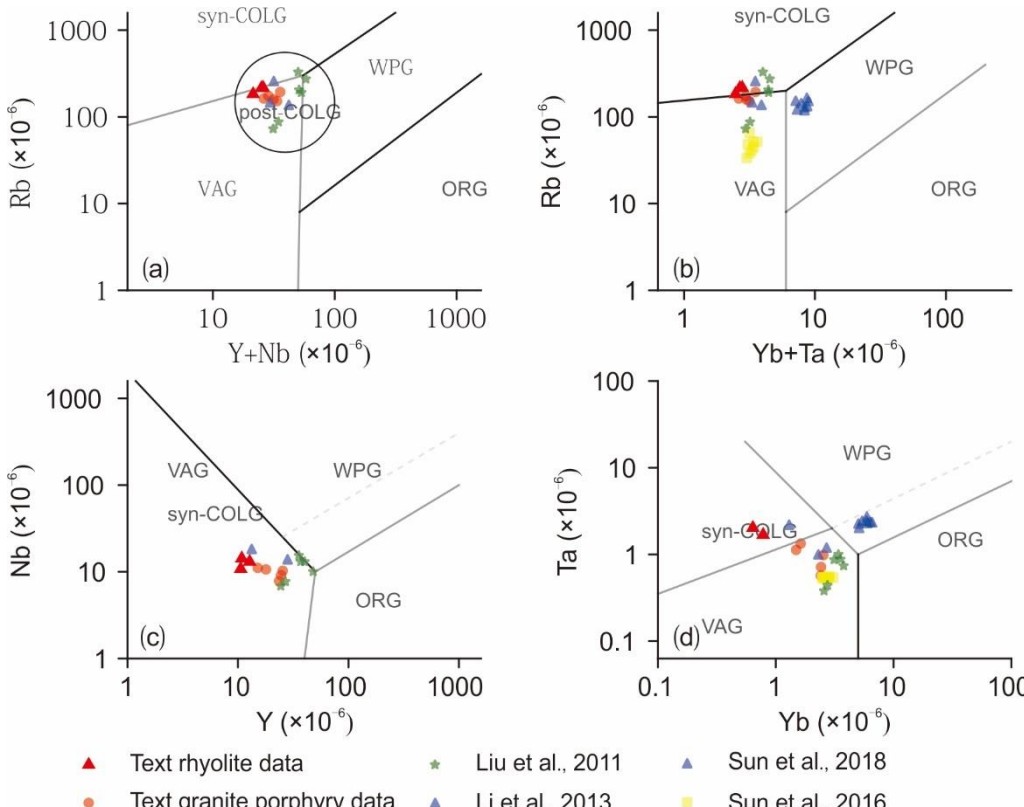

**Figure 10.** Tectonic discrimination diagrams for magmatic rocks (base image according to [82]). (**a**) Rb vs. (Y + Nb) diagram; (**b**) Rb vs. (Yb + Ta) diagram; (**c**) Nb vs. Y diagram; (**d**) Ta vs. Yb diagram. Syn-COLG, syn-collision granite; VAG, volcanic arc granite; WPG, within plate granite; ORG, ocean ridge granite; post-COLG, post-collision granite. The samples from Li et al. [17] and Sun et al. [44] are granite porphyries not far from the study area along the Ailaoshan orogenic belt. The samples from Liu et al. [54] and Sun et al. [43] are volcanic rocks not far from the study area along the Ailaoshan orogenic belt.

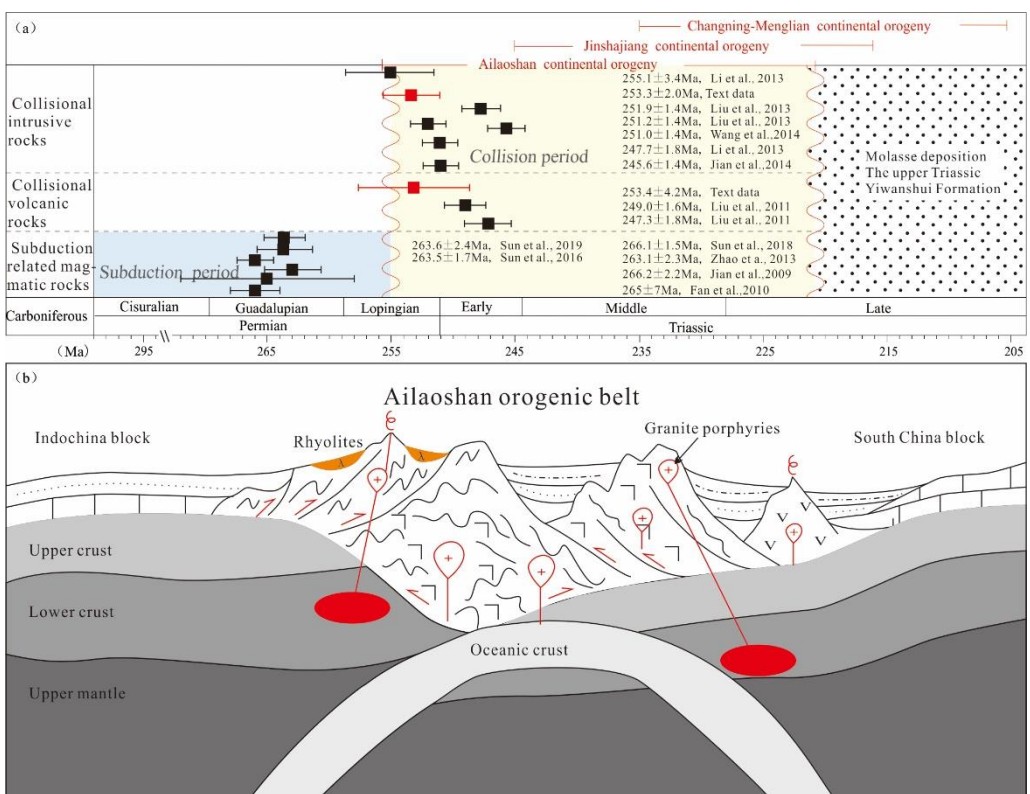

**Figure 11.** Schematic evolution diagram showing the collision time of the Ailaoshan orogenic belt based on magmatic ages (**a**) (modified by [17]) and a conceptual tectonic model of the Ailaoshan orogenic belt during the late Permian (**b**).

In summary, we hold the view that the Ailaoshan Ocean closed at about 255 Ma, and the rhyolite and granitic porphyry recorded a late Permian magmatic event in connection with the continental collision after the Ailaoshan Ocean basin closed. The evolution process of the rhyolites was as follows: at the end of the late Permian, the continuous convergence of the Indochina Block and the South China Block led to the continuous subduction and gradual extinction of the Ailaoshan Oceanic crust (Figure 11b). This process caused a crust thickening (for example, in the form of thrust–nappe deformation) and mantle material melting, and the resulting magma intruded into the bottom of the lower crust. Under the conditions of ultra-high temperature and pressure, partial melting of the crustal material occurred, forming the parental magma of rhyolite. Geochemical characteristics showed that the mantle-derived magma was not mixed into the magma, which might provide a heat source for the forming of crust-derived magma. Crust-derived magma rose to the shallow regions of the crust, crystallization differentiation occurred, and finally, it erupted to the surface. The evolution process of granite porphyries was as follows: influenced by the continuous convergence of the Indochina Block and the South China Block, mantle material melted and heated crustal material [17]. With the continuous interaction between mantle magma and crustal material, mixed magma was formed that was the primary magma of granite porphyry. The magma rose and emplaced along the weak part of the structure to the shallow area of the crust, undergoing crystallization differentiation, and finally, formed granite porphyry. According to Xu et al. [18], the U-Pb ages of detrital zircons from Late Triassic sedimentary rocks in the E margin of Indochina Block are concentrated in 240–260 Ma. The zircons in this age range have the characteristics of magmatic zircons, and the $\varepsilon$Hf (*t*) values of the zircons are mostly negative ($-15\sim0$). The U-Pb ages of detrital zircons from Late Triassic sedimentary rocks in the W margin of South China Block also have a peak age between 240 Ma and 260 Ma. The zircons at this age phase have the characteristics of magmatic zircons, while the $\varepsilon$Hf (*t*) values of the zircons are a mixture

(−10~10). Based on the above discussions, the *ε*Hf (*t*) values of the rhyolites indicated an affinity with the Indochina Block, and the Hf isotopic characteristics of the granite porphyries showed an affinity with the South China Block [18].

## 5. Conclusions

Based on the regional geology, geochemistry, and isotopic studies of zircons associated with rhyolite and granite porphyry in Mojiang and Zhenyuan gold deposits, the following conclusions were made:

(1) Both the rhyolite of Mojiang gold deposits and the granite porphyry of Zhenyuan gold deposits in the middle section of the Ailaoshan orogenic belt are strong peraluminous sub-alkaline series rocks, which were formed in a tectonic environment of continental collision.

(2) The weighted average age of the rhyolite was 253.4 ± 4.2 Ma, and the weighted average age of the granitic porphyry was 253.3 ± 2.0 Ma, both of which were formed in the late Permian.

(3) The rhyolite magma source area was mainly derived from ancient crustal materials (Ailaoshan Group); The magma source area of granite porphyry was mainly derived from depleted mantle and ancient crustal material (Ailaoshan Group).

(4) The ages of the rhyolite and granitic porphyry limited the time of the ocean–continent transition and, consequently, the closure of the Ailaoshan Ocean and the initiation of the continental collision.

**Supplementary Materials:** The following supporting information can be downloaded at: https://www.mdpi.com/article/10.3390/min12050652/s1, Table S1: Major element (wt%), REE element (ppm), and trace element (ppm) compositions of magmatic rocks; Table S2: LA-ICP-MS zircon U-Pb data of the rhyolite and the granite porphyry; Table S3: LA-ICP-MS Lu-Hf analysis results of the rhyolite and the granite porphyry zircons [85–87].

**Author Contributions:** Conceptualization, Y.Z. and K.L.; methodology, K.L. and Y.W.; software, Y.D.; validation, Y.Z. and K.L.; formal analysis, Y.Z. and Y.W.; investigation, Y.Z. and Y.W.; resources, D.Z. and X.M.; data curation, Z.Z. and T.Y.; writing—original draft preparation, Y.Z.; writing—review and editing, K.L. and Y.W.; visualization, Y.D.; supervision, D.Z. and X.M.; project administration, D.Z.; funding acquisition, Y.Z. and K.L. All authors have read and agreed to the published version of the manuscript.

**Funding:** This study was supported by the fundamental research funds of CGS Research (JKY202004, JKY202011) and the project of China Geological Survey (DD20221677-2, DD20201165).

**Data Availability Statement:** Not applicable.

**Acknowledgments:** We are grateful for the assistance during fieldwork provided by Junping Wang, and Hengyong Lei from the Yunnan Gold Co., Ltd., Yunnan, China. The authors also thank Qiuye Yu of Changchun Institute of Technology and Lele Han of China University of Geosciences (Beijing) for the computer mapping work.

**Conflicts of Interest:** The authors declare no conflict of interest.

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
