# Peer review of "Geological Significance of Late Permian Magmatic Rocks in the Middle Section of the Ailaoshan Orogenic Belt, SW China: Constraints from Petrology, Geochemistry and Geochronology"

_minerals, doi:10.3390/min12050652_

Round 1
Reviewer 1 Report
Dear authors, I strongly encourage you to improve the English of the paper and adapt it for an international audience. The use of obsolete terms and complex wording prevents a proper understanding of the text. Please keep in mind this paper will be read by an international audience that, perhaps, is not familiar with the geology of SW China.
Moreover, I suggest you improve your efforts to use clear geological vocabulary and explain the geological background and the geological relationships between the Permian rhyolitic and granodioritic units with their hosts. I often had to scroll up and down to understand what did you mean. I also have to note that there are concepts that are not synonyms and require precision (i.e. orogen and tectonic domain, tectonic belt). This advice goes also for stratigraphical name and capitalization: rock is not equivalent to unit, Formation or Group. Please revise the International Stratigraphic Code.
Please find attached my observations across the text.
Figures must be revised.

Reviewer 2 Report
My review is as follows:
(1) I made some corrections in the text of manuscript which can be overtaken by by authors (see manuscript),
(2) I think it is better that the manuscript can be read by native speaker in English language because there are too much repetitions in the manuscript
(3) The authors must know if they write Fig. oder Figure in the manuscript. I wrote all these as Figure.

Reviewer 3 Report
Dear Editor/Authors
As indicated in the title, the authors deal with "petrogenesis of rhyolite and granite related to two ore deposits" and include geochemicsal data, isotope geochemistry, and geochronology. The paper probably has the potential to be published in this special issue but lots of work should be done before making this a good paper for an international journal like Minerals. I list my major concerns below and hope these would be helpful for the authors to revise this draft. Although English is not my mother tongue, I think that some part of the text need revision for English. Some words wrote in the two form, e.g., later/early or Later/Early, lower/upper or Lower/Upper, fault or Fault, formation or Formation.
1) Geochemical data indicate that intrusions are fall in the granodiorite field. But in the text, the authors talk about granite. In addition, the granite samples in this study have more different geochemical characteristics than those are presented from previous literatures. For example, A/Nk and A/CNK values calculated wrong. The CaO values are very low and can not increase significant value of A/NK than A/CNK.
2) Another issue is related to U-Pb dating. The MSWD of samples are 4.3 and 2.4; too high, indicating that the spot analyses included in the age determination may not be of the same group, and is not acceptable. In addition, I do not know why they dated granite again. The Zhenyuan granites are dated by Li et al. (2013) and their ages are similar.
3) Discussion section: In the rock formation age subsection, the authors write about the results again and I do not see any discussion. This does not make sense. Although, subsection petrogenesis and source area is more reasonable, but the authors could be used more geochemical diagrams for discrimination of source as well as granite type. Discussions on tectonic significance subsection is more focused on geodynamic history of district and lack of good relation between results and tectonic settings. Lines 562 to 594 must be move into introduction section and more focused on tectonic settings of study district with more discussion.
Regards
Round 2
Reviewer 1 Report
Sample coordinates are still missing
English needs once more a full revision. The use of synonyms will make the text more engaging (i.e., the word composed is used 6 times in 11 consecutive lines). Revise the tenses. A native English speaker MUST revise the text to make not only more engaging but also clearer.
In my first review, I pointed out that it was necessary to give extra details (some of them basic, such as location) of the rocks that were considered for the geochemical comparison (see figure 4). In this second version the authors added “The volcanic rock data with similar geological significance in the region were collected and sorted to classify rock types and distinguish aluminium saturation degree [43, 54].” citing Sun et al (2018) and Liu et al. (2011). Are those of Sun et al and Liu et al also rhyolites? Both articles are in Chinese (only English abstract), therefore not easy for an international audience. In the case of the intrusive rocks, the comparison is made with Sun et al. (2016), Liu et al. (2011) and He et al. (2003). I could not find any comment regarding these extra samples.
I insist with my previous demand.
Once more, in section 3.3 Results, samples are NOT dropped, fall or are deposited in/on a field, range or region!
There is a problem with the modal mineral proportion in the rhyolites and the chemical analysis, see L314-317: 1/7 ratio phenocryst/matrix and 40-50% of Plagioclase in matrix contradicts the low CaO and Na2O wt% and high K2O. Please explain.
Explain what do you mean in L343-344 “The degree of differentiation of light and heavy rare earths is high.” Compared to what? Do you mean HREE/LREE? Or between certain samples?
Avoid unnecessary repetitions such as “The ratio of light to heavy rare earths LREE/HREE is 6.90–13.4, and (La/Yb)N is 6.53–14.7. The HREEs are depleted, while LREEs are enriched.” Everytime you use “enriched or depleted” it must be clear what are you comparing to.
L373 “The zircons had euhedral forms, most of which were euhedral crystals. Part of the crystal edges and apexes were eroded, showing a subcircular shape,” -please use the present tense, not simple past- Are they euhedral or subhedral? Did you find any isotopical difference between those subrounded crystals and the euhedral grains?
L372 How many points were analysed in each sample? All them were considered?
L384 When you refer to “(…) the tectonic thermal event of Early Paleozoic”, it is mandatory to add a reference, otherwise it is an ad-hoc statemnt. I suggest replacing “the” with “a” and adding “age” after Early Paleozoic.
Figure 6. Two figures in the analytical error seems an excess to me. Leave one figure after the comma and please add it to the age value. Be consistent showing the same figures after the comma in the text and the figure 6. In figure 6b and 6d add a colour (different to black) to the weighted average line.
The numbers and text in figures 6a and 6c are really hard to see.
L411. Again, I pointed this out in version 1. Including two figures after comma in the TDM model ages is simply unrealistic(!). Calculate the error of this determination and you will found out why.
L412. TDMc vs TDM, please say a word about which type of model age calculation are you considering for your final analysis (single stage or two-stage) and why did you chose a average crust as second stage in the last case?
Figure 7/ 8. As in figure 6, please explain where are located in the Ailaoshan orogenic belt the samples from Liu et al. (2013), Li et al. (2013) and Sun et al. (2016) that you are using for a comparison. Once more, Liu et al. (2013) is written in Chinese, unaccesible for most of the scientific community therefore I strongly suggest you give extra details about these rocks (type of rock, age, sampling area, etc). Perhaps you can include this background information as an appendix. Was this database recalculated using the same Lu/Hf constants and normalizing factors (such as DM model)?
Replace alterated (obsolete) with “altered” thoughout the text.
L418-421 Do not repeat in the text information that can be seen in the table. “Lu-Hf isotopic data of 12 zircons from sample LWZ-02 (granite porphyry)were shown in Table 3. The zircon 176Lu/177Hf ratios are between 0.001381 and 0.002685. 418 176Yb/177Hf ratios range from 0.036415 to 0.065853. 176Hf/177Hf ratios varie from 0.282593 to 419 0.282741. fLu/Hf ratios range from -0.92 to -0.96, and (…)”. Showing the fLu/Hf is not very common in papers dealing Hf zircon data, could you please indicate what are the accepted value range for this parameter and what is it used for?
L430 “Rock forming period” is not an engaging title. I believe that in this section you must discuss the background of your geochronological results, in turn your text is a repetition of the results section (see L 434-442). I STRONGLY SUGGEST to build a proper discussion. Another suggestion, enhance the relevance of your age in the Mojiang gol deposit, reinforce the idea that it is the first time this deposit is dated (is it ?) in L438-440. Start the paragraph L441-449 explaining that your secong age dating is from another gold mine and state that it coincides (between errors) with those of Li et al. 2013 . I wonder if Li et al 2013 also sampled from dikes or from a sill; I can not tell because there is no indication about these samples and Li et al 2013 is not accesible to me.
The description of where the sample was collected (L442-449) must be moved to the sampling and methods section.
L430 Revise the English. These sentences are not sound at all. "Technician in the mine proposed that the rocks were formed in Jurassic period” please revise, seems like this is an opinion rather than justified statement. Moreover, if you say “in the mine”, it seems like they might change their opinion when they are “out of the mine”.
L452 L453 Once more, unnecessary repetitions. What do you mean with “the rock(…) has a high differentiation index (DI) between 88.49 and (…), highly differentiated granite <25(…), indicating that rock had experienced a high degree of crystallization differentiation” ? I fully understand the DI but what is this “<25”? Avoid two figures in DI and remove “differentiation index” cause the acronym was explained earlier
Thereis a very relevant topic that is not discussed in section 4.2 “Petrogenesis and area characterization” that is the role of the hydrothermal alteration (see the Rb/Sr ratio) in these rocks. Before going forward add a word about this and evaluate their degree of alteration in order to strenghten your conclusions (please check Ohta and Arai 2007, weathering index or some other index that help you build a case) . Area characterization should not be part of the discussion.
More comments in the pdf file.
L 523 Section 4.2 finishes with “Combined with petrogeochemistry and zircon Hf isotope characteristics, we proposed that the source rock of granite porphyries in Zhenyuan gold deposit might be metamorphic sandstones of the Ailaoshan Group, and partly influenced by mantle-derived materials.” This is hypothesis, how can you test it ? Is Hf or Nd-Sr data from these sandstones available? Which would be the reaction or model behind the melting of this sandstones? It should be ideal to build a background behind this proposal.
Discussion is not enganging at all and the results are not discussed but merely presented in context with those of other authors. I suggest to re think and build proper arguments based on the data presented herein.
L604-606 Explain why your TDMc values show an affinity with those of the North China craton. Where is this affinity explained across the text? I suggest adding up NCCraton data to figures 7 or 8.
L606-608 again “The evolution process of granite porphyries were as follows: influenced by the Indochina tectonic event, mantle material melted and heated crustal material”. Please discuss this statement and show your arguments to support this statement.

Reviewer 3 Report
Dear Editor/Authors
This revised manuscript has been improved, however, some problems are not solved yet. The paper still needs minor revision:
1) About granite rocks (petrography and geochemistry): The authors say "Maybe there's something different here, and I think this is because different samples have different degrees of alteration.
It needs to clarify differences. Higher degrees of alteration are not suitable for geochemical diagrams. Also, I have a problem with this sentence in the abstract.
The granite porphyries are high potassic calc-alkaline subalkaline series, which is similar to granodiorites in composition. Granite and granodiorite have different mineralogy and whole-rock geochemistry. In addition, Figure 3f does not show clear granite mineralogy and must be changed.
2) I am not convinced about subsection 4.1. It also explains the results and is not a true discussion.
3) Although the authors have corrected grammatical errors, it is still essential to polish the English.
Minor comments.
Figures 1 and 2: Please provide a color legend for Figure 1 also. By comparing scales, the location of Figure 2 in Figure1 must be changed.
line 176: What do you mean vein rocks? Dikes?
line 252: The phenocryst/matrix ration for rhyolite and granite. Check it, and also check the mineralogy.
Regards
Round 3
Reviewer 1 Report
Dear authors, now the paper has a better shape.
L440 It is not you who suggests that these ages are younger. The ages of Li et al (2013) are between 247 and 255 Ma (according to Figure 1), this is not younger than yours (~250 Ma). Yours are within the age range. I suggest mentioning the values offered by Li et al (2013) in the text and discuss them in the context of your new ages, otherwise the reader must go to Li´s paper and (again) it is in Chinese. What type of rocks did Li et al dated? same/ different? Perhaps this information can help you building the discussion.
In my last review, I asked to shortly explain to the audience whats the fHf value means and include (perhaps in the appendix) the formula used to calculate the TDM2 and TDMC. Is it possible?
Regarding Point#23, your response was:" Thanks for good suggestion! According to Xu et al. (2019), the εHf (t) values of samples
from the Indochina block are mostly negative (-15~0), and the εHf (t) values of samples from the
South China block are mixture (-10~10). Therefore, we suggest that the εHf (t) values of the rhyolites
indicate the affinity with the Indochina block and Hf isotopic characteristics of the granite porphyries
show the affinity with the South China block." I am not fully convinced with this response given that the manuscript does not mention which rocks or ages of the Indochina and SC blocks were considered. Consider that Xi et al. obtained the Hf data of detrital zircon ages. This is one of the highlights of your work and I think you should reinforce your arguments in this direction.
I hope that my comments would help you, cheers.
